# A Machine Learning-Based Method for Detecting Liver Fibrosis

**DOI:** 10.3390/diagnostics13182952

**Published:** 2023-09-14

**Authors:** Miguel Suárez, Raquel Martínez, Ana María Torres, Antonio Ramón, Pilar Blasco, Jorge Mateo

**Affiliations:** 1Gastroenterology Department, Virgen de la Luz Hospital, 16002 Cuenca, Spain; 2Medical Analysis Expert Group, Institute of Technology, Universidad de Castilla-La Mancha, 16071 Cuenca, Spain; 3Medical Analysis Expert Group, Instituto de Investigación Sanitaria de Castilla-La Mancha (IDISCAM), 45071 Toledo, Spain; 4Department of Pharmacy, General University Hospital, 46014 Valencia, Spain

**Keywords:** artificial intelligence, liver fibrosis, machine learning, cholecystectomy

## Abstract

Cholecystectomy and Metabolic-associated steatotic liver disease (MASLD) are prevalent conditions in gastroenterology, frequently co-occurring in clinical practice. Cholecystectomy has been shown to have metabolic consequences, sharing similar pathological mechanisms with MASLD. A database of MASLD patients who underwent cholecystectomy was analysed. This study aimed to develop a tool to identify the risk of liver fibrosis after cholecystectomy. For this purpose, the extreme gradient boosting (XGB) algorithm was used to construct an effective predictive model. The factors associated with a better predictive method were platelet level, followed by dyslipidaemia and type-2 diabetes (T2DM). Compared to other ML methods, our proposed method, XGB, achieved higher accuracy values. The XGB method had the highest balanced accuracy (93.16%). XGB outperformed KNN in accuracy (93.16% vs. 84.45%) and AUC (0.92 vs. 0.84). These results demonstrate that the proposed XGB method can be used as an automatic diagnostic aid for MASLD patients based on machine-learning techniques.

## 1. Introduction

Metabolic-associated steatotic liver disease (MASLD) is recent nomenclature provided for non-alcoholic fatty liver disease (NAFLD) [1]. This revised definition aims to modernize the classical definition of NAFLD, seeking to reflect the disease’s heterogeneity while relaxing restrictive alcohol consumption criteria and emphasizing the underlying metabolic dysfunction in these patients [1,2,3]. MASLD is the most common chronic hepatic disease, affecting approximately 25% of the world’s population. There are several geographical differences, and it is most frequent in China and the Middle East [4,5,6]. Several studies highlight the increase and significance of the global burden of disease related to chronic liver diseases, with the primary cause being the rise in the incidence and prevalence of MASLD [7,8,9]. In addition to cardiovascular risk and the progression of liver fibrosis, the importance of MASLD diagnosis lies in the possibility of developing hepatocellular carcinoma without cirrhosis. These mechanisms are still unknown [10,11,12].

Cholecystectomy is the most frequent surgery performed worldwide by general surgeons. It is estimated that, only in the U.S., between 750,000 and 1,000,000 procedures are performed annually [13]. Symptomatic cholelithiasis is the main indication of cholecystectomy, either by biliary colic or complicated gallstone disease (acute cholecystitis, cholangitis or acute biliary pancreatitis) [14]. It is also the main biliary manifestation of the metabolic syndrome, closely related with an MASLD diagnosis [15,16,17]. Most of the published studies have focused on complications after surgery [13,14,18,19,20], but there is also evidence that cholecystectomy has metabolic effects, especially in NAFLD patients [21,22,23]. Although pathophysiology is not well understood, it is believed that cholecystectomy produces a change in the regulation of biliary acid in the enterohepatic circulation by different pathways. Among these, the most important are those mediated by the Farnesoid X receptor (FXR) [15,24] and Fibroblast Growth Factor 19 (FGF19) [25,26]. Insulin resistance (IR) also plays an important role in this process [27,28,29]. These changes can trigger a MASLD de novo or the progression of liver fibrosis.

To improve this lack of knowledge, machine learning (ML) algorithms can be employed. Such systems are being implemented in the healthcare system to predict various types of diseases due to the multitude of variables that allow the extraction of patterns from the data using mathematical algorithms [30,31]. In the literature, several classification methods have been developed, including K-Nearest Neighbour (KNN) [32] and Bayesian Linear Discriminant Analysis (BLDA) [33]. Other systems that are also applied in classification include Support Vector Machines (SVM) [34,35], Decision Tree (DT) [36], and Adaptive Boosting (Adaboost) [37,38].

Due to the evidence on the risk of liver fibrosis secondary to cholecystectomy in MASLD patients, the aim of this study is to develop a system to help select which patients can safely undergo cholecystectomy without the expectation of long-term metabolic sequelae. If surgery is necessary, this tool will allow the detection of which patients will need to be followed up to monitor these potential consequences and to avoid surgery when the indication is doubtful. For this purpose, Machine Learning (ML) techniques will be employed to analyse the database and assess the different variables that can lead to hepatic fibrosis [39]. In this work, an algorithm based on eXtreme Gradient Boosting (XGB) has been proposed. This technique has been selected due to its properties, including scalability, parallel computing, and the incorporation of regularization techniques. The results obtained with the proposed method have demonstrated its superior performance, surpassing other ML methods, in accurately classifying the detection of liver fibrosis in patients previously diagnosed with MASLD and requiring cholecystectomy. Therefore, this system offers promising potential for improving the diagnostic process and enhancing the accuracy of the diagnosis for the proposed objective in this study.

## 2. Materials and Methods

### 2.1. Study Design and Population

A database from a multicentre retrospective cross-sectional study of 4 hospitals in Mexico that was performed from 2014 to 2020 was analysed for this study. This database can be found in the Harvard Dataverse [40]. These hospitals were: Medica Sur Clinic and Foundation, Central Military Hospital, General Hospital of Mexico “Dr Eduardo Liceaga” and Star Médica Hospital. This study was approved by the Ethics and Research Committee for Human Studies of the Medica Sur Clinic and Foundation.

The search was conducted in each hospital to obtain all patients of both genders who were older than 18 years old and had a diagnosis of MASLD or steatohepatitis. This diagnosis was carried out by liver biopsy or transient elastography (FibroScan). Histological diagnosis and staging of liver biopsies were performed using the validated score system validated by Kleiner et al. [41]. Non-invasive diagnosis by FibroScan was performed by an expert hepatologist of each medical centre using the specific cut-off validated for each equipment. These cut-offs were: F0–F1 (<6.2 kPa), F2 (6.2–7.8 kPa), F3 (8.2–12.5 kPa) and F4 (9.5–16.1 kPa) [42]. When the result of liver stiffness showed overlap between the two stages, the hepatologist decided to assign the patient to the closest fibrosis stage of the resultant value. Patients were divided into two groups to facilitate the staging: simple steatosis or mild fibrosis were assigned to F0–F1; and significant or advanced fibrosis or cirrhosis were assigned to F2–F4.

To avoid overlap between diseases, patients who had the diagnostic criteria of another chronic liver disease was excluded. This mainly includes chronic hepatitis B, a positive hepatitis C antigen, high alcohol consumption (defined as more than 3 drinks per day for men or more than 2 drinks per day for women) and iron overload (transferrin saturation > 50%).

To conduct this study, the patients were categorised into two distinct groups. The first group comprised patients who received a diagnosis of MASLD after cholecystectomy. The second group consisted of control patients.

### 2.2. Data Collection

Anthropometric and demographic data registered were age, gender, height (m), weight (kg), Body Mass Index (BMI) (kg/m^2^), hypertension, type 2 diabetes mellitus (T2DM) and dyslipidaemia (DL). Laboratory data were obtained at the moment of diagnosis or within 30 days of the diagnosis of MASLD. These variables were haemoglobin (g/dL), platelet count (10^3^/µL), the international normalized ratio (INR), glucose (mg/dL), albumin (g/dL), total cholesterol (mg/dL), low-density lipoprotein (LDL) (mg/dL), high-density lipoprotein (HDL) (mg/dL), triglycerides (TG) (mg/dL), total bilirubin (mg/dL), direct bilirubin (mg/dL), indirect bilirubin (mg/dL), alanine aminotransferase (ALT) (U/L), aspartate aminotransferase (AST) (U/L), alkaline phosphatase (ALP) (U/L), gamma-glutamyl transferase (GGT) (U/L) and lactate dehydrogenase (LDH) (U/L). These baseline characteristics are reflected in Table 1.

Non-invasive tests (NITs) were added to the original database to have other tools in addition to biopsy and FibroScan to assist in decision making. Fibrosis-4 (FIB-4) and AST to Platelet Ratio Index (APRI) were included in the study because they are 2 of the most widely used worldwide for liver fibrosis in MASLD patients [43].

### 2.3. Model Development

To perform the analysis, the XGB method was proposed as the reference system for developing the predictive model. XGB is a supervised learning method based on a decision tree and designed to be highly efficient, flexible, scalable, speed in execution and portable. XGB can handle a variety of data types. In addition, XGB provides parallel tree reinforcement that solves many data science problems quickly and accurately. XGB also includes second-order regularisation. This helps prevent overfitting, a common problem in machine learning, by improving model generalisation. For all these reasons, the XGB method was chosen to construct the proposed prediction model [36,44].

The supervised SVM algorithm [45] creates a hyperplane where the distance between two classes of data points is the maximum. This hyperplane is known as a decision boundary, which separates the classes of data points on either side of the plane. They are effective in high-dimensional spaces and efficient in memory management [45]. On the other hand, KNN [46] is a non-parametric algorithm that classifies data points based on their proximity to and association with other available data. This algorithm assumes that similar data points are close to each other. Accordingly, it finds the distance between data points, usually by Euclidean distance, and then assigns a category based on the most frequent category or average. The algorithm is simple and easy to apply. There is no need to create a model, set various parameters or make additional assumptions. The algorithm is multi-purpose [46]. Another supervised method is DT [47], which uses a collection of uncorrelated decision trees fused together to reduce variance and create more accurate data predictions. The hierarchical nature of a decision tree makes it easier to see which attributes are the most important; they are flexible algorithms and can handle any type of data (discrete or continuous) [47]. As for BLDA [48], it attempts to simultaneously model the variance between segments of different individuals and different segments of the same individual using multidimensional Gaussians. This technique searches for directions in space that have maximum discriminative power and attempts to assign a feature vector to the appropriate class to which it belongs, with the highest probability. The BLDA algorithm employs regularisation to avoid overfitting to high dimensional and noisy datasets [48]. Finally, logistic regression (LR) [49] is a classification algorithm used to predict the probability of a categorical dependent variable. In logistic regression, the dependent variable is a binary variable. Logistic regression is one of the simplest and most widely used Machine Learning algorithms for two-class classification. It is easy to implement and can be used as a baseline for any binary classification problem. It describes and estimates the relationship between a binary dependent variable and the independent variables [49].

In the ML algorithms implemented in this study, the different hyperparameters of each one of them have been adjusted. In the SVM method, a Gaussian kernel function was chosen with the parameters C = 1.2, sigma = 0.5, numerical tolerance = 0.001 and iteration limit = 100. For BLDA, a Bayesian kernel was adopted to optimize the application of the algorithm. When referring to the DT method, the parameters employed for adjustment were tree, maximum number of splits = 20, learning rate = 0.1 and number of learners = 50. In the KNN algorithm, the distance metric is Euclidean, and it uses 25 neighbours. Finally, our proposed method, XGB, was optimized by using the parameters adjusted eta = 0.20, minimum chil weight = 1, gamma = 0.3, alpha = 0.6, maximum depth = 7, lambda = 0.3, col sample by tree = 0.7 and maximum delta step = 3. Machine Learning Toolbox and MatLab Statistical (The MathWorks, Natick, Massachsetts; MatLab 2023) were used to design the models.

Figure 1 represents how the process was carried out. For the evaluation, 5-fold cross-validation was used to validate algorithm performance. For each fold, 70% of the patients were used to train, and the other 30% of patients were used for testing and validation. Patient data were only used in one group (train or test) to avoid the possibility of being used in both groups at the same time. Once the database was completed, the training and validation models of ML methods were performed. The study was repeated 100 times to obtain the mean values and standard deviation. This process avoids the noise of the sample, allowing statistically significant results to be achieved.

## 3. Results

This section describes the results obtained using the available data for training and validation to identify which of the variables collected were most significant for the purpose of the study. The performance of the proposed system, XGB, was compared with other ML methods usually used by the scientific community.

Initially, 407 patients meeting the diagnostic criteria for MASLD mentioned above were found. A total of 196 were excluded for not meeting the inclusion criteria.

Figure 2 summarizes the importance of the main variables of the predictive model. The Y-axis represents the weight and significance of each variable within the predictive model. Platelet count was the most important variable, followed by dyslipidaemia and T2DM as the main variables. BMI and hypertension were the following variables with similar importance, but they were far from those previously mentioned. As a peculiarity, low HDL levels and above-normal bilirubin levels should also be considered. FIB-4 obtained a high result within the variables under study, which means that it was an accurate NIT for evaluating liver fibrosis in these patients.

Table 2 displays the results of balanced accuracy, recall, F1 score and Kappa. As can be observed in this table, the proposed method XGB obtained a balanced accuracy of 93.16, which was 8.71% higher than KNN, and a recall of 93.25 was 8.63% higher for this variable in relation to KNN, the closest method to XGB. The same applied to all other variables. The proposed algorithm XGB performed better than the other ML methods. The method that achieved lower classification accuracy was based on LR, with a value of 75.64. This represented a difference of 17.52% in favour of the proposed system.

Table 3 shows the values of specificity, Area Under de Curve (AUC), Matthews Correlation Coefficient (MCC) and Degenerated Younden’s Index (DYI) of analysed and proposed methods (SVM, BLDA, LR, DT, KNN and XGB). MCC was one of the most reliable statistical indices. This score only achieved a high score if the prediction was performed correctly in the four categories of the confusion matrix (true positives, false positives, true negatives and false negatives). For this parameter, XGB obtained a value of 84.41%, which represented an improvement of 9.53% over the second method, KNN. XGB also obtained a specificity of 93.60% and an AUC of 0.92. Comparing these results with the method that was closer to our proposed system, XGB achieved 8.72% higher values for specificity and 8% for AUC compared to KNN. As can be appreciated, the same occurred with the rest of the parameters analysed in the table. XGB significantly improved the results of DT, BLDA and SVM systems, in addition to LR. This means that, for predicting liver fibrosis after cholecystectomy, the proposed method XGB was the most appropriate system to implement this tool.

Figure 3 represents the comparative of all receiver operating characteristics (ROC) of the different ML methods with respect to XGB, the proposed method. These curves were the result of representing the sensitivity and specificity. As can be seen, the proposed method displayed the largest area under the curve (0.92), followed by the KNN method (0.84). This larger area for the XGB method means that it had a more accurate prediction, which allows a better identification of MASLD patients at risk of hepatic fibrosis after cholecystectomy.

To jointly represent all metrics of the proposed method XGB with the different ML methods compared, a radar plot was performed (Figure 4). The left side of the figure represents the training phase, while the figure on the right side represents the test phase. The larger the area of the circle of the test set, the better the prediction method will be. As can be observed in this case, the XGB algorithm presented similar results in both phases. This means that the method did not overtrain or overfit. Therefore, the algorithm had good predictive performance, and it had the capability to generalize. As shown, the rest of the ML systems obtained a lower area, which implies that they were less reliable for the purpose of the study.

## 4. Discussion

Cholecystectomy and MASLD are two of the most common medical conditions when we refer to the gastrointestinal tract. Although the studies published on both topics are not usually linked to each other, there is evidence that cholecystectomy may worsen the course of MASLD [50,51,52]. Cholecystectomy has also been considered an independent risk factor for MASLD [53].

The Hispanic population is one of the world’s most prevalent populations for NAFLD, with an estimated rate of 29%. When referring to Mexican individuals, this rate increases to 33% [54]. When these prevalence data are updated to include the definition of MASLD, the prevalence increases to 41.3%, one of the highest in the world [55]. This problem is not limited to the adult population but also to childhood and adolescents [56]. All this is closely related to a higher number of patients with T2DM, metabolic syndrome and obesity, in addition to a higher genetic risk related mainly with the presence of polymorphism PNPLA3 (Patatin like phospholipase domain-containing protein 3). This gene is associated with the severity of liver fibrosis in MASLD patients, and it is more frequent in the Hispanic population [57].

Cholelithiasis is also a highly prevalent disease worldwide, with an estimated ratio between 8 and 11% of the adult population. This prevalence varies according to geographic area, with Mexico having a prevalence between 5 and 15% within Latin America [58]. Gallstone disease and MASLD are frequently associated. Roesch-Dietlen et al. published a recent study in the Mexican population in which about 71% of patients with cholelithiasis had histological alterations compatible with MASLD [59]. It is estimated that about 69,000 surgeries are performed annually in this country [60].

The pathophysiology of this relationship is not well understood, but it seems to share some pathways, such as insulin resistance and metabolic syndrome [61]. Although metabolic syndrome is defined as the presence of glucose intolerance, IR, central obesity, hypertension and dyslipidaemia [62], the presence of cholesterol gallstones could be considered another part of this syndrome because they share risk factors, pathogenesis and the prevalence of gallstones is higher in this population [15].

Bile is the principal way of excreting excessive cholesterol by bile acids (BA) and phospholipids. BA also regulates metabolism function interacting with some nuclear receptors such as farnesoid X receptor (FXR), pregnane X receptor (PXR), vitamin D receptor (VDR) and G-protein coupled bile acid receptor-1 (also known as TGR5). This implies that BA has an important function in energy regulation, glucose and lipid metabolism, inflammation and gut microbiome, among others [15,24,63]

Another important pathway is mediated by fibroblast growth factor 19 (FGF19). After removing the gallbladder, bile is continuously secreted into the small intestine, making a faster enterohepatic circulation and exposing the liver to a higher amount of bile acids that previously were stored in the gallbladder. The gallbladder mucosa makes it endocrinological function by interacting with the bile, making changes in its composition. The loss of this function after cholecystectomy is related to the production of FGF19, which is also linked with FXR [50,64]. BA stimulates FXR, causing an increase in FGF19 levels. When bile acids reach the terminal ileum, they are absorbed by enterocytes at this level, stimulating the FXR-mediated secretion of FGF19 into the portal circulation, causing negative feedback to the gallbladder. When there is no gallbladder, this feedback is absent, so there is an increase of bile acid synthesis and a disturbance on metabolic homeostasis, increasing the risk of developing MASLD. These findings were confirmed in mice, although the complete mechanism is not well understood [65,66].

Insulin resistance plays a very important role between cholecystectomy and MASLD. IR is a main risk factor in the development and progression of MASLD, and it is determinant for gallstone formation [62,67]. It increases biliary secretion of cholesterol and reduces BA synthesis, leading to a lithogenic bile composition [68]. As the gallbladder mucosa has its role in insulin regulation, cholecystectomy is associated with increased manifestations of IR due to dysregulation of these mechanisms. This has been shown in some studies, where people after cholecystectomy have an increase in fat levels, apolipoprotein B, insulin and in the Homeostatic Model Assessment for Insulin Resistance (HOMA-IR) index [64,69]. The worsening of metabolic syndrome is linked to an aggravation of the MASLD condition and a high risk of progression in hepatic fibrosis.

These complex mechanisms, in which cholecystectomy produces a decrease in the elimination of excess cholesterol and exposing the liver to higher concentrations of BA, in addition to other pathways, can lead to the onset or progression of liver fibrosis in MASLD patients [53].

In this study, it is concluded that platelets are the most important risk factor that should be taken into account in MASLD patients for liver fibrosis when cholecystectomy is performed. This may be explained because low platelet levels can be an early sign of advanced liver fibrosis or cirrhosis, which has not been investigated yet [70]. Special caution should be considered when dealing with elderly patients or those with poor control of DL, T2DM, hypertension or elevated BMI, as can be seen in the results of the study. DL is the second risk factor associated with hepatic fibrosis after cholecystectomy, followed by T2DM, BMI and hypertension, which had a similar value in the predictive model. The explanation for these results can be justified by the excess cholesterol and BA to which the liver is exposed after cholecystectomy [53].

As initially discussed, it was decided to add two of the most used NIT that could be calculated with the available data. This decision was based on the need to have a tool capable of assisting in decision making, especially prior to performing surgery. NIT can help to detect undiagnosed patients with advanced liver fibrosis and monitor their evolution in relation to potential complications secondary to liver fibrosis [71,72]. As shown in the results, FIB-4 was the most reliable NIT, so it can be useful in deciding to perform cholecystectomy, especially when there are doubts about the indication [73].

A review of the published literature related to the objective of the study was conducted, with a special focus on ML. Only one similar study was found. However, it employed the MAFLD definition [40]. In this study, long-standing cholecystectomy, defined as a period exceeding 6 months, patients were at risk of advanced liver fibrosis, especially when they presented an age over 50 years and dyslipidaemia. When the search was extended to NAFLD definition, some contradictory studies were observed, but most of them considered cholecystectomy to be a risk factor for liver fibrosis. Xie et al. [52] performed a study including almost 5000 patients, in which a positive correlation was concluded between cholecystectomy and liver fibrosis. Kichloo et al. [74] conducted a study in hospitalized patients in which it was obtained as a result that cholecystectomy and gallstone disease may be risk factors for NAFLD. Cortés et al. [64] conducted an interesting study evaluating the effect of cholecystectomy on hepatic fat accumulation and IR in non-obese Hispanic patients; it was obtained as a conclusion that cholecystectomy is a risk factor of metabolic syndrome-associated complications, particularly NAFLD. Rodríguez-Antonio et al. [16] also performed an observational study in which they emphasized the importance of early detection of risk factors and potential complications in patients diagnosed with MASLD who undergo cholecystectomy. They consider medical follow-up necessary after surgery to monitor liver function and prevent long-term complications. Lyu et al. [75] in their systematic review mentioned as a main conclusion that cholecystectomy should be taken into account as a potential risk factor for developing de-novo NAFLD.

As can be seen in the results, the proposed system XGB achieved the best results in all the parameters analysed, almost all of them above 90%. These results were clearly superior to all the other ML algorithms evaluated, with a difference close to 9% for all these parameters. LR demonstrated poorer results compared to the different ML methods employed in the study. This can be attributed to the fact that ML techniques can be more efficient and accurate than traditional statistical analyses conducted by logistic regressions, particularly when the sample size is small [74]. The proposed method XGB confirms its reliability for automatic classification of the aim of the study. It also presented a similar performance in the radar plot between the training and test phases. These similarities are due to the fact that the system achieves an optimal point of training without overtraining and overfitting. For this reason, this method provides a high capability of generation, which means that every time there is a new entry, the system produces a correct exit. It is also a method that presents high scalability and execution speed, so it can be helpful in decision-making in daily clinical practice.

## 5. Conclusions

In conclusion, cholecystectomy in patients with MASLD should be carefully considered, particularly in those with low platelet levels, poorly controlled dyslipidaemia, and type 2 diabetes mellitus. Special caution is warranted in cases of elevated BMI and uncontrolled hypertension, as these factors further increase the risk of liver fibrosis. The use of FIB-4 as a non-invasive tool can be valuable in the decision-making process for determining the suitability of cholecystectomy or its contraindication.

The proposed XGB algorithm proved to be the most effective in identifying the main risk factors for liver fibrosis after cholecystectomy in patients with MASLD. The other ML methods compared were less accurate. The superior performance of XGB and its scalability make it a promising tool to aid clinical decision-making.

By taking these findings into account, clinicians can better assess the risk of liver fibrosis in MASLD patients undergoing cholecystectomy and make informed decisions regarding surgical interventions to optimize patient outcomes and long-term liver health.

## Figures and Tables

**Figure 1 diagnostics-13-02952-f001:**
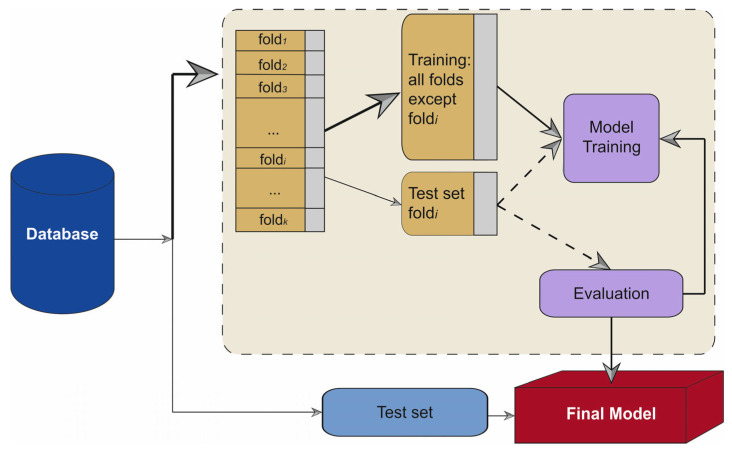
Diagram of how the machine learning method was performed.

**Figure 2 diagnostics-13-02952-f002:**
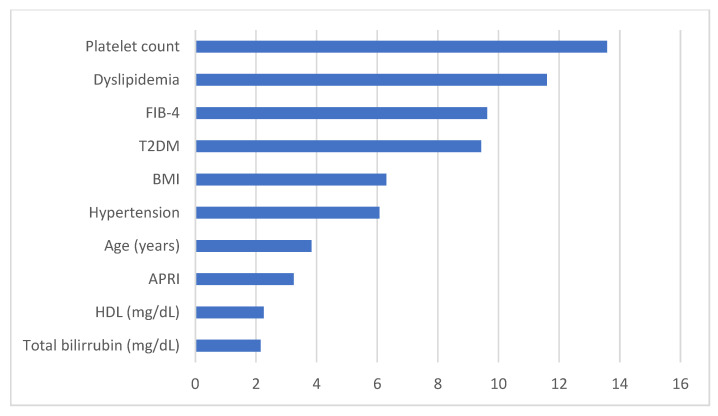
Representation of the most important variables and their value in the predictive model. Abbreviations. FIB-4: Fibrosis-4; T2DM: Type-2 Diabetes Mellitus; BMI: Body Mass Index; HDL: High-Density Lipoprotein cholesterol.

**Figure 3 diagnostics-13-02952-f003:**
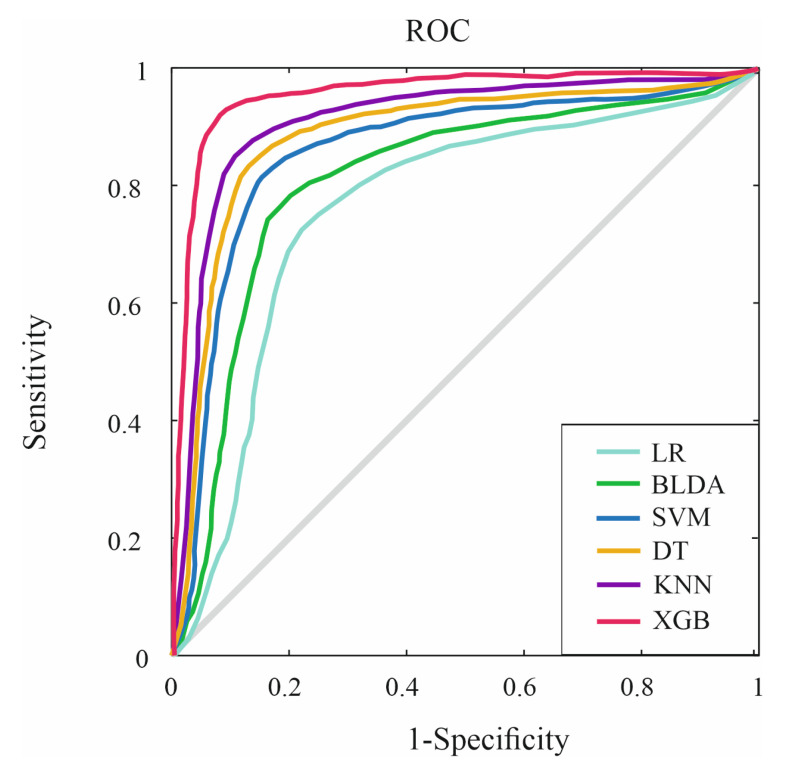
Representation of ROC curve of all compared methods. Abbreviations. ROC: Receiver Operating Characteristic; BLDA: Bayesian Linear Discriminate Analysis; SVM: Support-Vector Machine; LR: logistic regression; DT: Decision Tree; KNN: K-Nearest Neighbour; XGB: eXtreme Gradient Boost.

**Figure 4 diagnostics-13-02952-f004:**
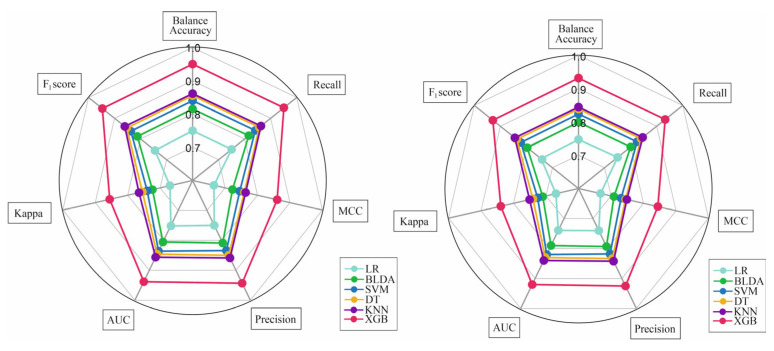
Radar plot of all compared methods. The train phase is represented on the left side of the image, while the test phase is drawn on the right. Abbreviations. BLDA: Bayesian Linear Discriminate Analysis; SVM: Support-Vector Machine; LR: logistic regression; DT: Decision Tree; KNN: K-Nearest Neighbour; XGB: eXtreme Gradient Boost; AUC: Area Under the Curve; MCC: Matthews Correlation Coefficient.

**Table 1 diagnostics-13-02952-t001:** Summary of baseline participant data. Data are also presented by study groups. BMI: Body Mass Index; ALT: alanine aminotransferase; AST: aspartate aminotransferase; ALP: alkaline phosphatase; GGT: gamma-glutamyl transferase; LDH: lactate dehydrogenase; LDL: Low-density lipoprotein; HDL; High-density lipoprotein; INR: International Normalized Ratio; FIB-4: Fibrosis-4; APRI: AST to Platelet Ratio Index.

	Global Population (Mean and Standard Deviation)	Patients with a Prior History of Cholecystectomy. (Mean and Standard Deviation)	Control Patients. (Mean and Standard Deviation)
Sample (*n*)	211	70	141
Age (years)	49.06 ± 15.15	53.15 ± 13.19	47.03 ± 15.69
BMI (Kg/m^2^)	29.19 ± 5.48	30.54 ± 5.37	28.52 ± 5.42
Gender	Men: 58 (27.49%)	Men: 18 (25.7%)	Men: 40 (28.4%)
	Women: 153 (72.51%)	Women: 52 (74.3%)	Women: 101 (71.6%)
Hypertension	Yes: 48 (22.75%)	Yes: 26 (37.1%)	Yes: 22 (15.6%)
	No: 163 (77.25%)	No: 44 (62.9%)	No: 119 (84.4%)
Type 2 diabetes mellitus	Yes: 57 (27.01%)	Yes: 29 (41.4%)	Yes: 28 (19.9%)
	No: 154 (72.99%)	No: 41 (58.6%)	No: 113 (80.1%)
Dyslipidemia	Yes: 26 (12.32%)	Yes: 19 (27.1%)	Yes: 7 (5%)
	No: 185 (87.68%)	No: 51 (72.9%)	No: 134 (95%)
Total bilirubin (mg/dL)	1.4 ± 1.68	1.27 ± 1.65	1.46 ± 1.69
ALT (U/L)	82.63 ± 143.63	57.51 ± 71.46	95.09 ± 167.20
AST (U/L)	77.81 ± 151.36	77.46 ± 118.58	77.99 ± 165.64
ALP (U/L)	135.97 ± 104.78	121.59 ± 72.23	143.72 ± 117.24
GGT (U/L)	95.31 ± 87.64	107.67 ± 114.14	88.65 ± 68.95
LDH (U/L)	244.42 ± 112.03	197.36 ± 101.35	269.76 ± 109.62
Hemoglobin (g/dL)	13.94 ± 1.88	13.64 ± 1.69	14.09 ± 1.95
Platelet count (10^3^/dL)	256.61 ± 97.24	236.23 ± 110.19	266.72 ± 88.82
Cholesterol (mg/dL)	188.17 ± 52.09	185.89 ± 49.23	189.4 ± 53.71
LDL (mg/dL)	102.23 ± 32.54	105.77 ± 32.67	100.33 ± 32.44
HDL (mg/dL)	39.6 ± 8.47	43.21 ± 9.49	37.65 ± 7.18
Triglycerides (mg/dL)	160.77 ± 77.37	164.56 ± 78.3	158.73 ± 77.09
Glucose (mg/dL)	116.22 ± 67.76	115.37 ± 54.21	116.64 ± 69.59
Albumin (mg/dL)	3.96 ± 0.66	4.08 ± 0.57	3.9 ± 0.69
INR	1.07 ± 0.17	1.07 ± 0.17	1.08 ± 0.17
FIB-4	2.11 ± 3.35	3 ± 4.92	1.66 ± 2.07
APRI	2.11 ± 1.53	1.08 ± 1.57	0.89 ± 1.52

**Table 2 diagnostics-13-02952-t002:** The table presents the balanced accuracy, recall, F1 score, and kappa values of the machine learning models studied and the proposed method. Abbreviations. SVM: Support-Vector Machine; BLDA: Bayesian Linear Discriminate Analysis; LR: logistic regression; DT: Decision Tree; KNN: K-Nearest Neighbour; XGB: eXtreme Gradient Boost.

Methods	Balanced Accuracy (%)	Recall	F1 Score	Kappa
SVM	82.39 ± 0.78	82.48 ± 0.73	82.14 ± 0.75	72.57 ± 0.74
BLDA	79.92 ± 0.93	80.02 ± 0.91	79.67 ± 0.92	71.00 ± 0.93
LR	75.64 ± 0.68	75.86 ± 0.73	75.58 ± 0.67	66.54 ± 0.64
DT	83.89 ± 0.72	83.99 ± 0.67	83.68 ± 0.68	74.02 ± 0.69
KNN	84.45 ± 0.65	84.62 ± 0.62	84.43 ± 0.64	75.05 ± 0.63
XGB	93.16 ± 0.53	93.25 ± 0.49	92.87 ± 0.50	83.91 ± 0.45

**Table 3 diagnostics-13-02952-t003:** The table shows the values of specificity, AUC, MCC and DYI values of the machine learning models studied and the proposed method. Abbreviations. AUC: Area Under the Curve; MCC: Matthews Correlation Coefficient; DYI: Degenerated Younden’s Index; SVM: Support-Vector Machine; BLDA: Bayesian Linear Discriminate Analysis; LR: logistic regression; DT: Decision Tree; KNN: K-Nearest Neighbour; XGB: eXtreme Gradient Boost.

Methods	Specificity	AUC	MCC	DYI
SVM	82.29 ± 0.77	0.82 ± 0.02	73.10 ± 0.74	82.37 ± 0.75
BLDA	79.82 ± 0.94	0.79 ± 0.02	70.91 ± 0.87	79.91 ± 0.92
LR	75.23 ± 0.65	0.75 ± 0.02	66.05 ± 0.67	75.47 ± 0.69
DT	83.80 ± 0.73	0.83 ± 0.02	74.50 ± 0.68	83.89 ± 0.71
KNN	84.37 ± 0.67	0.84 ± 0.01	74.88 ± 0.64	84.45 ± 0.63
XGB	93.06 ± 0.52	0.92 ± 0.01	84.41 ± 0.45	93.13 ± 0.50

## Data Availability

This database can be found at the Harvard Dataverse.

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
