# Peer review of "A Machine Learning-Based Method for Detecting Liver Fibrosis"

_diagnostics, 2023, doi:10.3390/diagnostics13182952_

Round 1

Reviewer 1 Report

Liver disease, specifically steatohepatitis and fibrosis, is a topical subject for research. The manuscript compares the effectiveness of different machine learning methods for building a predictive model.

One notices some inconsistencies between the article's title and content when looking at the manuscript. Still, the main goal is to compare different machine learning algorithms.

I wonder why the authors chose Kleiner's method of evaluating biopsies. Brunt's original classification provides more substantial information to the practicing clinical hepatologist.

Information about the organization of the manuscript (lines 74-78) seems redundant, and I recommend deleting it.

A brief overview of the differences between the various machine learning techniques should be added.

In general, I believe that the article can be published after minor revision.

Author Response

Please find attached a file

Reviewer 2 Report

This paper compared and evaluated various machine learning models to evaluate the risk of liver fibrosis in MASLD patients who received cholecystectomy. The performance evaluation results for the optimal model were shown.

Please correct the following information.

Figure 2 shows the importance of the main variables of the predictive model, but it is necessary to indicate what the Y-axis means.

Also, in the data collection section, sample data for the actual collected data should be shown, and it is necessary to present the cutoff point for each major variable.

In addition, in this paper, the number of platelets was suggested as a major risk factor for the diagnosis of liver fibrosis, and we would like to present a quantified numerical value.

Author Response

Please find attached a file

Reviewer 3 Report

The authors have utilised many machine learning algorithms, there is no novelty in the proposed work. They have simply used the available algorithms. There is no significant contribution.

The authors have utilised many machine learning algorithms, there is no novelty in the proposed work. They have simply used the available algorithms. There is no significant contribution.

Author Response

Please find attached a file
